# TriG-NER: Triplet-Grid Framework
# for Discontinuous Named Entity Recognition

## Abstract

Discontinuous Named Entity Recognition (DNER) presents a challenging problem where entities may be scattered across multiple non-adjacent tokens, making traditional sequence labelling approaches inadequate. Existing methods predominantly rely on custom tagging schemes to handle these discontinuous entities, resulting in models tightly coupled to specific tagging strategies and lacking generalisability across diverse datasets. To address these challenges, we propose TriG-NER, a novel Triplet-Grid Framework that introduces a generalisable approach to learning robust token-level representations for discontinuous entity extraction. Our framework applies triplet loss at the token level, where similarity is defined by word pairs existing within the same entity, effectively pulling together similar and pushing apart dissimilar ones. This approach enhances entity boundary detection and reduces the dependency on specific tagging schemes by focusing on word-pair relationships within a flexible grid structure. We evaluate TriG-NER on three benchmark DNER datasets and demonstrate significant improvements over existing grid-based architectures. These results underscore our framework's effectiveness in capturing complex entity structures and its adaptability to various tagging schemes, setting a new benchmark for discontinuous entity extraction.

## CCS Concepts

• **Do Not Use This Code → Generate the Correct Terms for Your Paper**; *Generate the Correct Terms for Your Paper*; Generate the Correct Terms for Your Paper; Generate the Correct Terms for Your Paper.

## Keywords

Discontinuous Named Entity Recognition, Medical Named Entity Recognition, Medical Text Mining

**ACM Reference Format:**
Anonymous Author(s). 2018. TriG-NER: Triplet-Grid Framework for Discontinuous Named Entity Recognition. In *Proceedings of Make sure to enter the correct conference title from your rights confirmation emai (Conference acronym 'XX)*. ACM, New York, NY, USA, 14 pages. https://doi.org/XXXXXXX.XXXXXXX

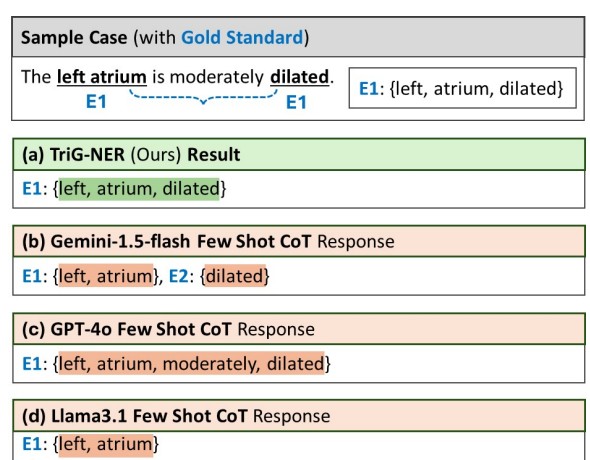

**Figure 1: A Case example involving discontinuous mentions with Gold Standard (a) Our proposed TriG-NER enables to perfectly extract the DNE (b,c,d) LLMs face challenges in DNER as those are primarily trained to capture continuous sequences of text, making it difficult for them to recognise entities split across discontinuous regions while maintaining coherence in prediction.**

## 1 Introduction

Named Entity Recognition (NER) is a fundamental task in natural language processing that involves identifying and categorising entities such as person names, locations, or temporal expressions within unstructured text. Traditionally, NER has been approached using sequential labelling techniques like the Begin-Inside-Outside (BIO) scheme, which assigns labels to each token in a sentence. However, while effective for contiguous entities, such schemes struggle to accurately capture discontinuous named entities whose mentions are interrupted by non-entity tokens due to their linear nature and inability to represent complex entity structures.

Recent research in Discontinuous Named Entity Recognition (DNER) has sought to address these limitations by introducing new tagging schemes and model architectures. These include extensions of the BIO scheme like BIOHD [28], span-based methods [34], and grid-based tagging [36], which attempt to represent more complex entity boundaries and relationships. While these methods have shown improvements in extracting discontinuous entities, they often suffer from heavy reliance on task-specific tagging strategies. This makes them highly specialised, limiting their adaptability to new datasets and unseen entity types. Moreover, current solutions primarily focus on sample-based learning objectives, which do not fully capture the token-level dependencies critical for recognising scattered entities. Generative and large language models (LLMs)

like ChatGPT have also been explored for DNER, using sequence-to-sequence approaches to generate entity spans. However, these models, optimised for next-word prediction, are not inherently suited for the intricate nature of NER tasks, making them prone to generating incorrect spans and entity boundaries. Grid-tagging methods, on the other hand, have achieved state-of-the-art performance in DNER by modelling word-pair relationships. Nevertheless, they often lack a mechanism to differentiate between similar and dissimilar word-pair representations, particularly for discontinuous entities separated by non-entity tokens.

To address these challenges, we introduce **TriG-NER**, a Triplet-Grid Framework that leverages token-based triplet loss to learn fine-grained word-pair relationships for DNER. Unlike traditional triplet loss, which operates at the sample level by comparing entire sequences, our method applies triplet loss at the token level, where similarity is defined by word pairs co-occurring within the same entity. This approach enables the model to capture the local dependencies between tokens in discontinuous entities, ensuring that word pairs forming an entity are cohesively represented in the learned feature space. We also propose a grid-based triplet loss that models word-pair relationships within a flexible grid structure, where positive pairs represent tokens within the same entity, and negative pairs include word pairs disrupted by non-entity tokens. The main contributions of this paper are as follows:

**1. Token-based Triplet Loss for NER**: We introduce a novel token-based triplet loss that learns fine-grained token-level representations for discontinuous entity extraction, contrasting with existing methods that use sample-based triplet loss.

**2. Grid-based Triplet Loss Using Word-Pair Relationships**: We propose a grid-based triplet loss that defines word-pair similarity based on co-occurrence within the same entity, enhancing the model's ability to capture non-adjacent entity segments.

**3. Extensive Evaluations and Qualitative Analysis**: We perform extensive evaluations on three widely used DNER benchmark datasets and provide a qualitative analysis that demonstrate the effectiveness of our grid-based triplet framework over existing baselines and prompted large language models.

## 2 Related Works

### 2.1 Discontinuous Named Entity Recognition

Named entity extraction and recognition has traditionally been viewed as a sequence labelling task using the Begin-Inside-Outside (BIO) tags; however, this traditional approach fails for more complex entities such as discontinuous entities. Researchers have recently focused on improving discriminative discontinuous entity recognition through various tagging schemes and methods. Tang et al. (2015) [28] was the first to extend BIO sequential tagging to BIOHD to distinguish inter-entity boundaries, which subsequent studies [19, 29] followed. More recently, Corro (2024) [3] proposed a two-layer tagging scheme that uses ten tags; however, these methods fail to capture complex discontinuous entities and suffer from decoding ambiguity. Span-based methods [11, 14, 34] typically involve the identification of all candidate spans and the merging of disjoint spans. The two-step process, however, is vulnerable to error propagation and identifying all possible span candidates is

**Table 1: Comparison of NER schemes and losses in recent works in discontinuous named entity recognition.**

| DNER Models | Core Scheme | Loss |
|---|---|---|
| Corro (2024) [3] | Sequence Tagging | NLL |
| Wang et al. (2019) [34] | Span-based | NLL |
| Li et al. (2021) [14] | Span-based | NLL |
| Huang et al. (2023) [11] | Span-based | NLL |
| Mao et al. (2024) [17] | Span-based | BCE |
| Dai et al. (2020) [4] | Transition-based | - |
| Wang et al. (2021) [36] | Grid Tagging | CE |
| Li et al. (2022) [15] | Grid Tagging | NLL |
| Liu et al. (2022) [16] | Grid Tagging | CE |
| Fei et al. (2021) [6] | Seq2Seq | NLL |
| Yan et al. (2021) [39] | Seq2Seq | NLL |
| Zhang et al. (2022) [41] | Seq2Seq | - |
| Xia et al. (2023) [38] | Seq2Seq | MLE |
| Zhao et al. (2024) [42] | Prompting | - |
| Zhu et al. (2024) [43] | Prompting | - |
| **Ours** | Word-Pair Grid Tagging | Triplet |

resource-exhaustive. Other discriminative methods, such as hypergraphs [22, 33] and stack-and-buffer transitions [4], are also explored yet still suffer from error propagation. On the other hand, generative methods [6, 38, 39, 41], leverage sequence-to-sequence language models to directly generate entity spans and types that overcome the challenges presented by different complex entity structures. With the advent of ChatGPT, research in applying large language model (LLM) prompting to discontinuous NER has also seen increased attention [42, 43]. However, generative models are optimised for next-word prediction, not NER, predisposing it to incorrect biases.

Grid tagging [36], another discriminative method, has shown state-of-the-art performance through identifying spans using word pair tags defining word-pair relationships [15, 16]. However, grid tagging approaches are still constrained by their reliance on specific grid tag designs and decoding strategies. Moreover, they tend to treat word pairs independently, failing to capture the contextual relationships between word pairs that could enhance the recognition of discontinuous entities. This lack of dependency modelling between similar and dissimilar word pairs can result in the misclassification of complex, scattered entity spans. To address these limitations, we propose TriG-NER, a novel Triplet-Grid Framework that integrates token-based triplet loss with grid tagging to model fine-grained word-pair relationships. Unlike existing methods that treat word pairs in isolation, our approach leverages triplet loss to distinguish between similar and dissimilar word pairs, enhancing the model's ability to recognise non-adjacent entity segments.

### 2.2 Triplet Loss

Triplet loss [25] was introduced in the computer vision (CV) area in the field of facial recognition or reidentification [7, 20, 40] for deep metric learning by directly optimising image sample embeddings. Unlike contrastive loss, triplet loss takes three points - an anchor, a positive, and a negative - and ensures that the positive is closer to the anchor than the negative point by a certain margin. This

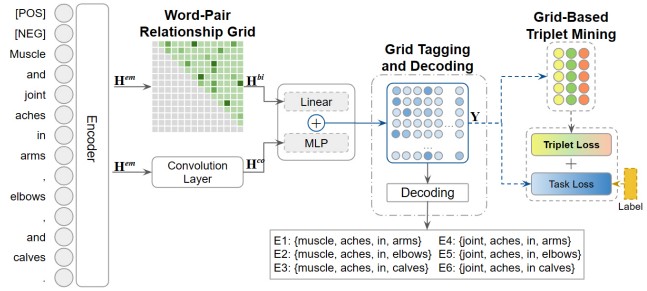

**Figure 2: Overall framework of the proposed TriG-NER**

optimisation effectively pulls together images belonging to the same person and pushes away seemingly similar images that do not share the same identity, producing a better feature space. As a result, triplet loss has seen wide adoption and a few variations in other CV fields, such as image segmentation [27], facial synthesis [35], 3D object retrieval [10], and medical image classification [9]. In the area of natural language processing (NLP), researchers have explored the use of triplet loss for text classification [18, 37], relation extraction [26], and spoken language understanding (SLU) [24, 32].

However, traditional triplet loss is typically employed at the sample level, where similarity is defined by class membership, which does not necessarily align with the needs of discontinuous entity extraction. Detecting discontinuous entities requires capturing local dependencies and boundary information within entities scattered across non-adjacent tokens. Our proposed framework addresses these limitations by introducing a grid-based, token-level triplet loss, where word-pair co-occurrence within the same entity defines similarity. This approach ensures that entity tokens are drawn closer together in the feature space, even when interrupted by non-entity tokens that may appear syntactically or semantically similar. To the best of our knowledge, no existing work has applied a grid-based, token-level triplet loss for discontinuous named entity recognition, making our approach a novel contribution to this field.

## 3 Methodology

In this study, we propose a new type of DNER architecture that utilises word-pair relationships in a grid structure, along with grid-based triplet mining to improve discontinuous entity extraction. Our framework builds on recent advances in grid tagging and word-to-word relation classification, introducing a novel combination of grid-based tag decoding and triplet loss mechanisms. This section provides an overview of a grid-based NER model, our newly proposed NER model with a word-pair relationship grid, grid tagging and decoding, and grid-based triplet loss.

### 3.1 Grid-based NER Models

Recent studies on Named Entity Recognition (NER) have explored using grid-based tagging schemes to improve discontinuous entity extraction, especially where traditional sequence tagging approaches like the Begin-Inside-Outside (BIO) scheme fall short. In grid-based models, the NER task is treated as a word-to-word relation classification problem, where a sequence input $X = \{x_1, x_2, ..., x_n\}$ of length $n$ is transformed into a grid output $\mathbf{Y} = \{y_{11}, y_{12}, ..., y_{nn}\} \in$

$\mathbb{R}^{n \times n \times c}$, where $c$ is the number of tag classes. Each element $y_{ij} \in \mathbb{R}^c$ represents the logits used to calculate the probability of a relationship between word $i$ and word $j$.

Grid-based NER models focus on word-pair relationships, where token pairs, rather than individual tokens, are labelled. This structure allows for representing complex, non-contiguous entity structures, making it a flexible method for DNER. Existing models such as those proposed by [15] and [36] have shown promising results by utilising these word-to-word grids, which map the relationships between tokens, allowing models to handle both contiguous and non-contiguous entities effectively. However, these models treat each word pair independently, which overlooks the inherent relationships between multiple word pairs that can exist within the same entity. This lack of dependency modelling between similar and dissimilar word pairs can result in misclassifications, particularly when dealing with complex, non-adjacent entity structures.

### 3.2 Word-Pair Relationship Grid

Hence, we address this limitation by introducing triplet loss at the word-pair level, which enables the model to explicitly learn the fine-grained distinctions between similar and dissimilar word pairs within the grid. To achieve this, we introduce a word-pair relationship grid to explicitly model the relationships between words within entities. The proposed word-pair relationships are treated as the primary feature for entity extraction, and the overall NER task is transformed into a word-pair classification problem.

The input sentence is first passed through an encoder layer, where we utilise pre-trained language models (PLMs) such as BERT [5], BioClinicalBERT [2], PharmBERT [31], and PubMedBERT [8]. These models generate contextualised word embeddings $\mathbf{H}^{em} \in \mathbb{R}^{n \times d}$, where $d$ is the embedding dimension. A bidirectional LSTM layer is then applied to capture sequential dependencies in the sentence. The embeddings are then passed through two distinct modules: a Convolution Layer and a Biaffine transformation. The Convolution Layer generates enhanced word-pair representations $\mathbf{H}^{co} \in \mathbb{R}^{n \times n \times d^{co}}$, where $d^{co}$ is the convolution dimension, while the Biaffine transformation computes word-pair relationships $\mathbf{H}^{bi} \in \mathbb{R}^{n \times n \times d^{bi}}$. These representations are combined in a Co-Predictor Layer, where a linear layer and an MLP map $\mathbf{H}^{bi}$ and $\mathbf{H}^{co}$ to tag relation logits $\mathbf{Y}^{bi}$ and $\mathbf{Y}^{co} \in \mathbb{R}^{n \times n \times c}$. The final grid tag logits are obtained by combining the two: $\mathbf{Y} = \mathbf{Y}^{bi} + \mathbf{Y}^{co}$.

### 3.3 Grid Tagging and Decoding

The grid tagging system classifies word-pair relationships using three tag classes: *None*, *Next-Neighboring-Word* (NNW), and *Tail-Head-Word* (THW). These classes define whether a word pair has no relationship, a neighbouring relationship within an entity, or represents the start and end of an entity, respectively. Once word-pair relationships are classified, the grid decoding process begins, which is crucial for discontinuous entity extraction. The system takes the final grid tag logits $\mathbf{Y}$ and decodes the predicted relationships into entity structures. By focusing on word pairs rather than individual tokens, the grid structure allows our model to flexibly identify discontinuous entity boundaries, which are common in complex entity recognition tasks. The grid tagging and decoding approach enables the model to handle non-contiguous entity spans

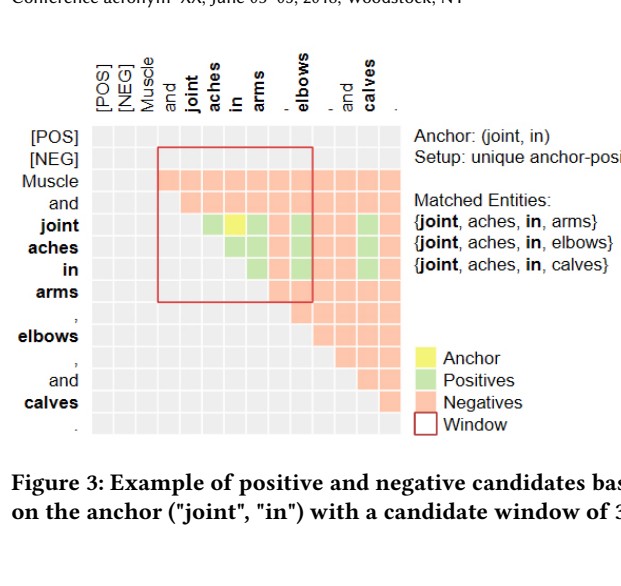

**Figure 3: Example of positive and negative candidates based on the anchor ("joint", "in") with a candidate window of 3.**

by considering the relationships between word pairs, making it robust against the limitations of sequential tagging schemes.

## 3.4 Grid-based Triplet Mining

*3.4.1 Preliminaries.* To further optimise the model's performance in capturing discontinuous entities, we introduce a grid-based triplet loss, which enables the model to learn distinctions between similar and dissimilar word pairs more effectively. Triplet loss is a metric learning objective that brings similar word pairs closer while pushing dissimilar pairs farther apart. The loss function is defined as $L_{triplet} = \sum max(f(a, p) - f(a, n) + m, 0)$ where $a$ is an anchor point, $p$ is a positive point similar to the anchor, $n$ is a negative point dissimilar to the anchor, $f$ is a distance function, and $m$ is a margin that ensures a minimum distance between negative pairs and positive pairs. We utilise Euclidean distance for our distance function. Our final loss combines the triplet loss with the task loss: $L_{final} = L_{task} + L_{triplet}$.

*3.4.2 Word-Pair Grid Implementation.* We extract our triplets from the word-pair grid representations in our framework. Unlike most sample-based triplet loss implementations that define similarity by sample classes, we define the similarity of our triplet elements based on their existence within entities. For the anchor candidates, we use word-pair grid points that exist in any entity. Each anchor candidate is then matched with positive and negative candidates. Positive candidates are word pairs that co-exist with the anchor in any entities, while negative candidates are word pairs that don't belong to any entity the anchor is a part of. We illustrate this candidate selection in Figure 3.

For special instances, we incorporate two special tokens [POS] and [NEG] at the start of each sample. These special instances include one-word entities and anchor points that do not have other positive or negative word pairs to match with. For the example in Figure 4, the sentence "Insomnia was constant ." with "Insomnia" as an entity uses $cell_{insomnia,[POS]}$ as the anchor point. Since no other positive point could be matched, $cell_{[POS],[POS]}$ is the only positive candidate. In cases where no negative candidates can be used, $cell_{[NEG],[NEG]}$ is used. We experiment with extracting our

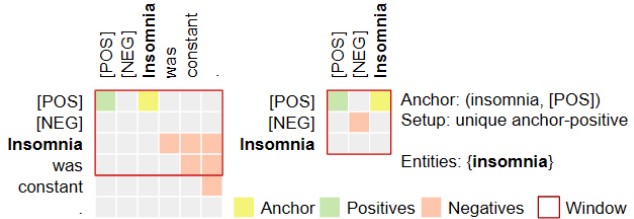

**Figure 4: Example of positive and negative candidates for one-word entities (left) and one-word samples (right).**

triplet representations from the Word-Pair Relationship Grid ($\mathbf{H}^{bi}$) or from the final output logits ($\mathbf{Y}$).

*3.4.3 Triplet Selection.* It is crucial to select valid triplets that violate the triplet constraint wherein the positive candidates are farther from the anchor than the negative candidates by a margin [25]. Since generating all possible anchor-positive-negative combinations not only exponentially increases computation time and resources needed but, more importantly, generates uninformative triplets that result in slower convergence during training, we utilise different online triplet selection methods illustrated in Figure 5.

(1) **Hard Negative (HN)** selection takes each anchor-positive combination and selects the closest negative candidate from the anchor.
(2) **Semi-hard Negative (SN)** selection takes each anchor-positive combination but, different from the hard negative, selects the negative candidate that is closest to the anchor but farther than the positive point within the set margin.
(3) **Centroid (CE)** takes the mean of all the positive candidates and the mean of all the negative candidates for each anchor as the positive and the negative points.
(4) **Negative Centroid (NC)** utilises all anchor-positive pairs but takes the mean of all the negative candidates as the negative point.

Due to the exponential increase of positive and negative candidates as the sample length increases, we further limit the positive and negative candidate selection by using a candidate window centred on the anchor and by specifically using unique anchor-positive pairs. The unique anchor-positive pair setup utilises only the top half triangle of the grid (Figure 3) where an anchor token pair $tp_1$ is paired with a positive candidate $tp_2$, but when $tp_2$ is set as an anchor, $tp_1$ will not be considered as a positive candidate anymore. This reduces possible redundant information that is not helpful for training while simultaneously reducing the number of triplets. A comparison of performance between unique and non-unique anchor-positive pairs is provided in Table 5.

## 4 Experimental Setup

### 4.1 Datasets

Following previous studies on discontinuous named entity recognition, we use three datasets in the biomedical domain to assess the performance of our proposed system. The CSIRO Adverse Drug Event Corpus (**CADEC**) [12] is a collection of medication consumer posts annotated for entity identification from the public

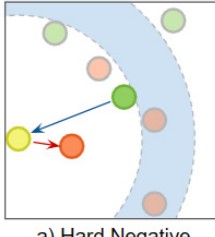 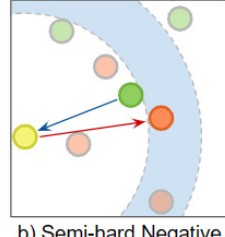 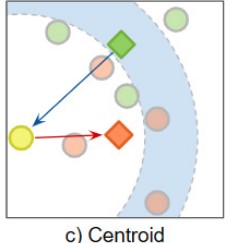 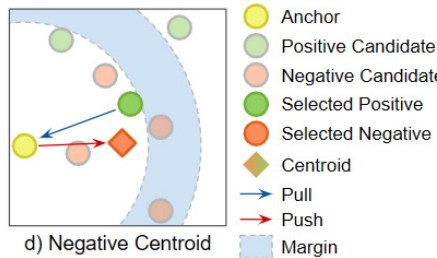

Figure 5: Triplet Mining Methods

Table 2: Data statistics

|  | CADEC | ShARe13 | ShARe14 |
|---|---|---|---|
| Total Sentences | 7,597 | 18,767 | 34,618 |
| Total Entities | 6,318 | 11,148 | 19,073 |
| Continuous Entities | 5,639 | 10,060 | 17,417 |
| - Percentage | 89.25% | 90.24% | 91.32% |
| - Number of tokens | 1-36 | 1-9 | 1-9 |
| Disc. Entities | 679 | 1,088 | 1,658 |
| - Percentage | 10.75% | 9.76% | 8.68% |
| - Number of tokens | 2-13 | 2-7 | 2-7 |
| - Start-End Distance | 3-20 | 3-23 | 3-23 |

forum AskAPatient. We follow previous literature and use only the adverse drug reaction (ADR) entities. **ShARe13** [23] and **ShARe14** [21] datasets are part of the Shared Annotated Resources used for the CLEF eHealth Challenge in 2013 and 2014, respectively. They consist of clinical reports annotated for the identification and normalisation of disease disorders. For all datasets, we use the sentence-based preprocessing script and dataset splits provided by Dai et al. [4] and convert the produced inline format to JSON[1] following Li et al. [15]. Table 2 shows each dataset's statistics.

### 4.2 Baselines and Metrics

We compare our framework with other DNER models. **MAC** [36] first introduced the grid tagging scheme with a segment extractor labelling relative token pairs using the BIS (begin, inside, continuous) scheme and an edge predictor which aligns entity bounds using the *head-to-head* (H2H) and *tail-to-tail* (T2T) tags. $\mathbf{W^2NER}$ [15] introduced a unified NER framework that identifies neighbouring word relationships between non-adjacent entity words using the tags *Next-Neighboring-Word* (NNW) and *Tail-Head-Word* (THW). **TOE** [16] improves upon the $W^2$NER's tagging scheme by adding *Previous-Neighboring-Word* (PNW) and *Head-Tail-Word* (HTW) and incorporating a *Tag Representation Embedding Module* (TREM). **Corro** [3] is a recent model attempting to improve sequence tagging for discontinuous entities through a two-layer tagging system using ten tags. For both $W^2$NER and TOE, we report reproduced results using the published code from each study. Following previous NER studies, we evaluate our framework through exact matching of entities using micro-F1, precision, and recall. We

[1]Script provided in the code repository.

further isolate the effect of our framework on discontinuous entities by reporting F1 scores for sentences with discontinuous entities and for discontinuous entities only (Table 3).

### 4.3 Implementation Details

We evaluate our framework using the established training, validation, and test splits by [4]. We list best-performing model setups for each dataset in the Appendix D. Each model is trained using the AdamW optimiser with a learning rate of 5e-4 for a maximum of 60 epochs and an early stop of 10 epochs. We take the best-performing model on the validation set based on the micro-F1 score. We use a batch size of 12, 6, and 6 for CADEC, ShARe13, and ShARe14, respectively. Our best setup for the CADEC dataset uses a fine-tuned BioBERT, while both ShARe datasets achieve better results with fine-tuned PubMedBERT. A comparison of PLMs is provided in Table 7. All models are trained using an NVIDIA RTX A4500.

## 5 Results

### 5.1 Overall Performance

A comprehensive evaluation of our framework compared to other studies is provided in Table 3. The results reflect the performance of our framework on the entire test set, as well as on discontinuous elements, with isolated evaluations on sentences containing at least one discontinuous entity (DiscSent) and on discontinuous entities exclusively (DiscEnt). Our framework demonstrates a clear improvement in both F1 score and precision over $W^2$NER, the best-performing baseline method. The ShARe14 dataset shows the most significant improvement in F1 score, with a 1.23% increase, reaching 82.54. Similarly, the CADEC and ShARe13 datasets show increases of 0.76% (73.43) and 1.06% (83.22), respectively. Furthermore, our framework outperforms the baseline models when focusing on discontinuous elements, with improvements of 0.79%, 0.63%, and 3.19% for DiscSent, and 3.98%, 2.68%, and 5.13% for DiscEnt across the CADEC, ShARe13, and ShARe14 datasets, respectively. Complete performance metrics may be found in Appendix A. These results underscore the strength of our TriG-NER framework in capturing the complexities of discontinuous entities by leveraging word-pair similarities and dissimilarities. By focusing on token-level relationships within a flexible grid structure, our approach demonstrates superior performance in both overall entity recognition and specifically in handling discontinuous elements, highlighting its adaptability and effectiveness compared to traditional methods.

**Table 3: Comparison of performance from our best-performing models for the overall datasets and for discontinuous elements, including sentences containing at least one discontinuous entity (DiscSent) and discontinuous entities only (DiscEnt). Bold indicates best scores while underline shows next best. [†] indicates replicated results.**

| | Overall | | | DiscSent | DiscEnt |
|---|---|---|---|---|---|
| **CADEC** | **F1** | **P** | **R** | **F1** | **F1** |
| MAC [36] | 71.50 | 70.50 | 72.50 | 69.80 | 44.40 |
| W$^2$NER[†] [15] | 72.67 | 72.02 | 73.33 | 69.25 | 45.78 |
| TOE[†] [16] | 72.24 | 74.28 | 70.30 | 67.98 | 40.00 |
| Corro [3] | 71.90 | - | - | - | 35.90 |
| Ours | **73.43** | 75.35 | 71.62 | **70.59** | **49.71** |
| **ShARe13** | **F1** | **P** | **R** | **F1** | **F1** |
| MAC [36] | 81.20 | 84.30 | 78.20 | 68.10 | 55.90 |
| W$^2$NER[†] [15] | 82.16 | 84.13 | 80.29 | 68.46 | 57.38 |
| TOE[†] [16] | 81.92 | 85.05 | 79.02 | 67.82 | 57.06 |
| Corro [3] | 82.00 | - | - | - | 52.10 |
| Ours | **83.22** | 86.44 | 80.24 | **69.09** | **60.06** |
| **ShARe14** | **F1** | **P** | **R** | **F1** | **F1** |
| MAC [36] | 81.30 | 78.20 | 84.70 | 69.70 | 54.10 |
| W$^2$NER[†] [15] | 81.31 | 78.93 | 83.84 | 63.08 | 52.70 |
| TOE[†] [16] | 80.67 | 78.67 | 82.78 | 61.04 | 49.29 |
| Corro [3] | 81.80 | - | - | - | 49.80 |
| Ours | **82.54** | 80.36 | 84.83 | **72.89** | **59.23** |

## 5.2 Triplet Selection

We evaluated the performance of our framework using various triplet selection methods and configuration setups. Table 4 shows the performance of our framework under the best-performing model setup for each selection method since different window sizes may affect each method's effectiveness. Among the four strategies, the Centroid strategy consistently shows promising results among the four selection strategies across all datasets, producing the best scores for overall CADEC and both subsets of ShARe13, while securing the second-best scores for the others. The Negative Centroid strategy also demonstrated encouraging outcomes, having the best score for overall ShARe14 and a competitive second-best for overall CADEC with only a 0.1% disadvantage. On the other hand, the Semi-Negative strategy showed a notably high score for the DiscEnt subset of CADEC. However, it sacrifices overall performance, which falls short of the baseline score, possibly signifying the benefits of a stricter negative candidate selection for the discontinuous entities in the dataset. Similarly, the Hard Negative follows the same trend for ShARe14. Nonetheless, we note that all our triplet selection methods, except Hard Negative, generally outperform and are competitive with the baseline model. This highlights the benefits of leveraging word-pair relationships through our grid-based triplet framework with careful consideration of triplet selection strategies.

In Table 5, we compare other design setups for our framework. Using unique anchor-positive pairs through only the top half of the grid sources generally shows superior performance compared

**Table 4: Comparison of different triplet selection methods based on the best-performing setup for each method. Bold indicates best scores while underline shows next best. [†] indicates replicated results from the baseline. HN: Hard Negative; SN: Semi-hard Negative; CE: Centroid; NC: Negative Centroid**

| Method | CADEC | | ShARe13 | | ShARe14 | |
|---|---|---|---|---|---|---|
| | Overall | DiscEnt | Overall | DiscEnt | Overall | DiscEnt |
| [15][†] | 72.67 | 45.75 | 82.16 | **57.38** | 81.31 | 52.70 |
| HN | 71.61 | 45.41 | 81.79 | 54.45 | 81.87 | **57.35** |
| SN | 72.21 | **49.35** | 82.56 | 56.30 | 82.19 | 53.79 |
| CE | **73.43** | 48.55 | **83.22** | 57.14 | 82.42 | 56.22 |
| NC | 73.33 | 46.75 | 82.43 | 56.22 | **82.54** | 54.40 |

**Table 5: Comparison of the anchor-positive pairing and triplet embedding source design setups. Bold indicates best scores while underline shows next best.**

| | Setup | CADEC | ShARe13 | ShARe14 |
|---|---|---|---|---|
| **Pairing** | Unique | **73.43** | **83.22** | **82.54** |
| | Non-unique | 71.73 | 81.82 | 82.09 |
| **Source** | Word-Pair Grid ($\mathbf{H}^{bi}$) | 71.22 | 81.19 | **82.54** |
| | Grid tag logits ($Y$) | **73.43** | **83.22** | 82.23 |

to using the entire grid. Utilising only half of the grid lessens uninformative and redundant triplets while also reducing the computational time and resources needed. To highlight the flexibility of our framework, which could be applied to any model with a grid-based component, we further analysed different triplet embedding sources for our framework. Directly applying the triplet loss on the grid tag logits ($Y$) shows noticeably better performance for CADEC and ShARe13. On the other hand, for ShARe14, the results for both sources are comparable, with a slight improvement from the Word-Pair Relationship Grid ($\mathbf{H}^{bi}$). These findings underscore the effectiveness and versatility of our framework in enhancing discontinuous entity extraction by incorporating word-pair relationships and optimising triplet selection strategies.

## 5.3 Window Size

Given the importance of selecting informative triplets for the triplet loss, we applied a window size centred on the anchor to restrict the positive and negative candidates. In this section, we evaluate the impact of different window sizes on the performance of our best model setups across each dataset. As shown in Table 6, implementing a window significantly improves our framework's performance compared to no window, though the optimal window size varies depending on the dataset. For example, the longer entities in the CADEC dataset benefit from larger window sizes. In contrast, both ShARe datasets achieve optimal performance with smaller window sizes, as the entities in these datasets range from 1 to 9 tokens in length. Removing the window altogether and allowing the framework to select triplets from the entire sequence grid introduces less informative triplets, leading to lower overall performance. Specifically, we observed an improvement of 1.94%

**Table 6: Comparison of different window sizes. Bold indicates best scores while underline shows next best.**

| Window Size | CADEC | ShARe13 | ShARe14 |
|---|---|---|---|
| None | 71.49 | 81.74 | 81.78 |
| 1 | 71.65 | 81.21 | 81.91 |
| 5 | 72.77 | 82.02 | **82.54** |
| 10 | 72.88 | **83.22** | 81.19 |
| 15 | 70.84 | 81.26 | 80.81 |
| 20 | 70.67 | 81.79 | 81.33 |
| 25 | **73.43** | 81.83 | 81.83 |

for CADEC, 1.48% for ShARe13, and 0.76% for ShARe14. Our results demonstrate the critical role of window size in enhancing the triplet selection process, ensuring that only the most relevant triplets are used to optimise the learning process. This highlights our framework's adaptability to various dataset characteristics, leading to consistent improvements in performance by effectively leveraging the word-pair relationships within a controlled selection window.

## 5.4 Encoder Language Models

We evaluated the performance of our framework with different pre-trained language models for the encoder, using the best-performing model setup for each dataset. Table 7 presents the results for four biomedical BERT variants, both with and without our grid-based triplet framework. Overall, BioBERT yields the best results for the CADEC dataset, while PubMedBERT outperforms others for both ShARe datasets. The application of our framework further enhances these scores by 0.93%, 1.22%, and 1.12%, respectively, demonstrating that our framework effectively captures local dependencies via the word-pair triplet implementation. Additionally, our framework consistently improves the performance of most PLMs tested, with the exception of BioClinicalBERT for CADEC and ShARe14. In Table 8, we present the performance improvements achieved by finetuning the pre-trained language models using a next-word prediction task for each dataset. As expected, finetuning enhances the scores across the board, with more pronounced improvements observed in the ShARe datasets, likely due to the specialised clinical terminology in those datasets compared to the more natural language used in online forums like CADEC.

## 5.5 Qualitative Analysis

In this section, we demonstrate the effectiveness of our word-pair grid-based triplet framework through a qualitative analysis of the extracted entities, comparing the results with those of the best-performing baseline model and LLMs, such as Gemini 1.5-flash [30] and GPT-4o [1]. We trained and fine-tuned both our model and the replicated baseline model using tokenised sentences as direct inputs, while the LLMs were not fine-tuned and were provided with task-specific prompts that described the task, input, and expected output format. For few-shot prompts, we included two examples from the training data. Table 10 presents a case study based on a CADEC sample, with additional case studies and prompt templates available in Appendix E and Appendix F.

**Table 7: Comparison of different language models used in the encoder with and without our triplet framework based on the best-performing setup for each dataset. Bold indicates the overall best scores for each dataset while an underline shows the better score regarding the application of our framework.**

| PLM | TriG-NER | CADEC | ShARe13 | ShARe14 |
|---|---|---|---|---|
| BioBERT [13] | × | 72.50 | 80.25 | 80.75 |
| | ✓ | **73.43** | 80.72 | 80.79 |
| BioClinicalBERT [2] | × | 71.49 | 81.78 | 81.00 |
| | ✓ | 71.42 | 81.89 | 80.27 |
| PharmBERT [31] | × | 70.78 | 80.25 | 80.00 |
| | ✓ | 71.90 | 80.39 | 81.11 |
| PubMedBERT [8] | × | 70.19 | 82.00 | 81.42 |
| | ✓ | 71.39 | **83.22** | **82.54** |

**Table 8: Comparison of performance from finetuning the pre-trained language models for the encoder layer. Bold indicates best scores while underline shows next best.**

| Setup | CADEC | ShARe13 | ShARe14 |
|---|---|---|---|
| Pretrained | 72.96 | 81.35 | 80.38 |
| Finetuned | **73.43** | **83.22** | **82.54** |

While our framework uses the same tags as $W^2$NER, it goes further by leveraging word-pair relationships to accurately recognise multiple non-adjacent entity segments within the input text. In contrast, $W^2$NER processes word pairs in isolation, which limits its ability to recognise entities with more than two disjoint spans, such as "Pain in my lower legs" and "cramping in my lower legs", indexed as [0, 3, 4, 7, 8] and [2, 3, 4, 7, 8], respectively. Furthermore, $W^2$NER struggles to detect uncommon, domain-specific terms and abbreviations, particularly when the entity consists of just one word. For example, in Figure F4, our framework successfully extracts the entity "PFO", which stands for "Patent Foramen Ovale", despite the presence of other domain-specific terms. By contrast, $W^2$NER incorrectly extracts "MV", which in this context likely refers to "mitral valve", but is not a disorder.

With LLMs' recent success and popularity for general language generation tasks, we evaluate their performance in extracting entity indexes through zero-shot and few-shot chain-of-thought (CoT) prompting. Because LLMs are optimised for next-word prediction, these models are prone to alignment and indexing problems where, despite clear instructions, the indexes returned do not correspond to the entity words identified. We found that explicitly including the entity words in the return format prompt helps partially but does not entirely resolve the problem. For instance, in Figure F2, the entity words "loss of range of motion" are correctly identified; however, the indexes provided are one or two positions off. In some cases, the number of words identified does not equate to the number of indexes returned, such as "{'entity': 'loss of range of motion', 'index': [32, 36], 'type': 'ADR'}". Furthermore, LLMs fail to extract discontinuous entities most of the time. In Table 10, both Gemini and GPT-4o completely missed the overlapping continuous and discontinuous entities in the sample despite identifying relevant parts

**Table 9: Comparison of triplet loss margins. Bold indicates best scores while underline shows next best.**

| Margin | CADEC | ShARe13 | ShARe14 |
|--------|-------|---------|---------|
| 0.1    | 72.58 | 81.88 | 82.16 |
| 0.5    | 71.72 | 81.78 | 81.86 |
| 1      | **73.43** | **83.22** | **82.54** |
| 1.5    | 71.76 | 81.70 | 82.18 |
| 2      | 71.41 | 82.16 | 80.93 |

such as "Pain and cramping", "hands", and "lower legs". They cannot effectively split and combine disjoint spans to form discontinuous entities such as "Pain in my hands" and "Pain in my lower legs". GPT-4o Few-shot CoT goes as far as returning the whole input instead of associating the relevant spans together. Lastly, LLMs are prone to extracting entities unrelated to the entity type provided. For instance, body parts such as "hands" and "lower legs" in Table 10 and medical procedures such as "CABG" (coronary artery bypass graft surgery) are separately identified as ADRs and Disorders, respectively.

While general LLMs have shown significant progress, they still face limitations in specialised tasks like discontinuous entity extraction, unless meticulously designed prompts are used. Trained models continue to outperform current attempts to adopt LLMs for biomedical NER [42]. Our framework, which enhances current trainable DNER models by using token-level, grid-based triplets to account for the similarity and dissimilarity of word pairs, delivers superior performance, especially in handling complex discontinuous entity recognition.

## 5.6 Hyperparameter Testing

We conducted further tests to investigate the impact of different triplet loss margins on the best-performing setup for each dataset. As shown in Table 9, using a margin of 1 consistently delivers superior performance across all datasets. In contrast, using a margin of 2 results in a significant performance drop for CADEC and ShARe14, with reductions of 2.02 and 1.61 points, respectively. Similarly, a margin of 1.5 causes a decline of 1.06 points for ShARe13. These results highlight the sensitivity of our framework to the triplet loss margin and the importance of carefully tuning this hyperparameter. The consistently strong performance with a margin of 1 underscores the robustness of our triplet-based model in capturing word-pair relationships, ensuring optimal performance across different datasets.

## 6 Conclusion

In this paper, we introduced TriG-NER, a novel Triplet-Grid Framework designed to improve the extraction of discontinuous named entities by leveraging token-level triplet loss and word-pair relationships. By modelling token pairs within a flexible grid structure, our framework overcomes the limitations of existing tagging schemes, which often struggle to generalise across different datasets.

We evaluated TriG-NER on three benchmark DNER datasets, demonstrating significant improvements over state-of-the-art grid-based architectures. The results validate the effectiveness of our

**Table 10: Case study for CADEC comparing the entity extraction results from trained models using our TriG-NER framework, a baseline model, and LLMs employing zero-shot and few-shot chain-of-thought (CoT) prompt engineering. The table compares how each method identifies discontinuous entities within a sample sentence from the CADEC. The models trained with our framework demonstrate more accurate entity recognition, especially for non-adjacent entity segments. Prompt templates for the LLMs are provided in Appendix E. Green highlight indicates correctly identified entities. Red highlight indicates otherwise.**

| Input |
|-------|
| ['Pain', 'and', 'cramping', 'in', 'my', 'hands', 'and', 'lower', 'legs', '.'] |

| Gold Standard |
|-------|
| {'entity': 'Pain in my hands', 'index': [0, 3, 4, 5], 'type': 'ADR'}, 
 {'entity': 'Pain in my lower legs', 'index': [0, 3, 4, 7, 8], 'type': 'ADR'}, 
 {'entity': 'cramping in my lower legs', 'index': [2, 3, 4, 7, 8], 'type': 'ADR'}, 
 {'entity': 'cramping in my hands', 'index': [2, 3, 4, 5], 'type': 'ADR'} |

**Ours - 4/4 (100%)**

{'entity': 'Pain in my lower legs', 'index': [0, 3, 4, 7, 8], 'type': 'ADR'}, 
{'entity': 'Pain in my hands', 'index': [0, 3, 4, 5], 'type': 'ADR'}, 
{'entity': 'Pain in lower legs', 'index': [0, 3, 7, 8], 'type': 'ADR'}, 
{'entity': 'cramping in my hands', 'index': [2, 3, 4, 5], 'type': 'ADR'}, 
{'entity': 'cramping in my lower legs', 'index': [2, 3, 4, 7, 8], 'type': 'ADR'}, 
{'entity': 'cramping in lower legs', 'index': [2, 3, 7, 8], 'type': 'ADR'}

**W$^2$NER - 2/4 (50%)**

{'entity': 'Pain in my hands', 'index': [0, 3, 4, 5], 'type': 'ADR'}, 
{'entity': 'cramping in my hands', 'index': [2, 3, 4, 5], 'type': 'ADR'}

**Gemini Zero Shot CoT - 0/4 (0%)**

{'entity': 'Pain', 'index': [0], 'type': 'ADR'}, 
{'entity': 'cramping', 'index': [2], 'type': 'ADR'}, 
{'entity': 'hands', 'index': [5], 'type': 'ADR'}, 
{'entity': 'lower legs', 'index': [7, 8], 'type': 'ADR'}

**Gemini Few Shot CoT - 0/4 (0%)**

{'entity': 'Pain and cramping', 'index': [0, 1, 2], 'type': 'ADR'}, 
{'entity': 'hands', 'index': [5], 'type': 'ADR'}, 
{'entity': 'lower legs', 'index': [7, 8], 'type': 'ADR'}

**GPT-4o Zero Shot CoT - 0/4 (0%)**

{"entity": "Pain", "index": [0], "type": "ADR"}, 
{"entity": "cramping", "index": [2], "type": "ADR"}

**GPT-4o Few Shot CoT - 0/4 (0%)**

{"entity": "Pain", "index": [0], "type": "ADR"}, 
{"entity": "cramping", "index": [2], "type": "ADR"}, 
{"entity": "Pain and cramping", "index": [0, 1, 2], "type": "ADR"}, 
{"entity": "Pain and cramping in my hands", "index": [0, 1, 2, 3, 4, 5], "type": "ADR"}, 
{"entity": "Pain and cramping in my hands and lower legs", "index": [0, 1, 2, 3, 4, 5, 6, 7, 8], "type": "ADR"}

approach in capturing non-adjacent entity segments and underscore the framework's ability to adapt to various tagging schemes, setting a new standard for discontinuous entity extraction. Future work could explore integrating our framework with larger language models and expanding its application to other structured prediction tasks, such as relation extraction and event detection. We hope that our framework, with its innovative grid-based triplet approach, will inspire further research into developing generalisable methods for discontinuous named entity recognition in structured prediction.

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

## A  Comprehensive Metric Scores

We provide the F1, Precision, and Recall scores from our overall best-performing model in Table A1. In Table A2, we present the performance scores from the model setup that scores highest for the discontinuous entities only (DiscEnt). We observe significantly higher scores for discontinuous entities for the best DiscEnt model with 1.66%, 2.92%, and 4.83% for CADEC, ShARe13, and ShARe14, respectively. However, despite not having the best overall scores in our experiments, the best DiscEnt models still outperform all of the baselines for the CADEC and ShARe14 datasets and are comparable to our overall best model highlighting the ability of our framework to extract discontinuous entities through word-pair triplets.

**Table A1: Complete performance scores from the best-performing overall model for sentences with at least one discontinuous entity (DiscSent) and for discontinuous entities only (DiscEnt).**

| Dataset | DiscSent | | | DiscEnt | | |
|---|---|---|---|---|---|---|
| | F1 | P | R | F1 | P | R |
| CADEC | 70.54 | 75.52 | 66.18 | 48.55 | 53.16 | 44.68 |
| ShARe13 | 69.23 | 79.14 | 61.53 | 57.14 | 71.23 | 47.71 |
| ShARe14 | 64.82 | 65.64 | 64.01 | 54.40 | 60.96 | 49.12 |

**Table A2: Complete performance scores from the best-performing discontinuous entity model for the overall dataset, for sentences with at least one discontinuous entity (DiscSent), and for discontinuous entities only (DiscEnt).**

| | Overall | | | DiscSent | | | DiscEnt | | |
|---|---|---|---|---|---|---|---|---|---|
| | F1 | P | R | F1 | P | R | F1 | P | R |
| **CADEC** | 73.22 | 75.00 | 71.52 | 70.59 | 73.81 | 67.64 | 49.71 | 54.43 | 45.74 |
| **ShARe13** | 81.35 | 85.60 | 77.50 | 69.09 | 79.44 | 61.13 | 60.06 | 78.52 | 48.62 |
| **ShARe14** | 82.16 | 79.78 | 84.69 | 72.89 | 74.25 | 71.59 | 59.23 | 57.60 | 60.95 |

# B Token Gap Analysis

Figure B1 shows the difference in token gaps between CADEC, ShARe13, and ShARe14. CADEC generally shows shorter gaps between spans for discontinuous entities, while the ShARe datasets have wider gaps despite having shorter entities. These differences present unique challenges for extracting discontinuous entities in each dataset, highlighting the need for a flexible and adaptable solution like our proposed framework.

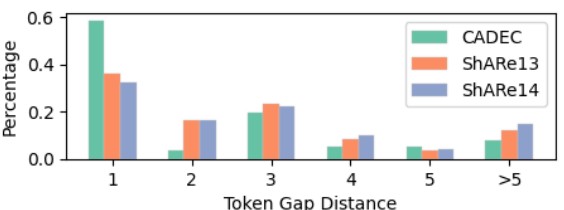

**Figure B1: Distribution of token gaps of discontinuous entities.**

# C Hyperparameter Study

We investigate the effect of different hyperparameter values on our best overall model. In Table C1, we test different learning rate values for the Adam optimiser and find that the optimal learning rate value for our framework is 5e-04.

**Table C1: Comparison of learning rates. Bold indicates best scores while underline shows next best.**

| Learning Rates | CADEC | ShARe13 | ShARe14 |
|---|---|---|---|
| 1e-03 | 72.40 | 81.00 | 81.56 |
| 5e-04 | **73.43** | **83.22** | **82.54** |
| 3e-04 | 71.68 | 81.72 | 82.08 |
| 2e-05 | 69.53 | 80.87 | 81.62 |

# D Best-found Parameter Setup

**Table D1: Parameter setup for the best model based on overall performance scores for each dataset.**

| Setting | CADEC | ShARe13 | ShARe14 |
|---|---|---|---|
| PLM | BioBERT | PubMedBERT | PubMedBERT |
| Window Size | 25 | 10 | 5 |
| Triplet Method | Centroid | Centroid | Neg. Centroid |
| Learning Rate | 5e-04 | 5e-04 | 5e-04 |
| Source | Grid Tag Logits | Grid Tag Logits | Word-Pair Grid |

# E Large Language Model Prompts

**Table E1: Prompt templates used for large language models.**

| Prompt Type | Content |
|---|---|
| Zero Shot CoT | "The task is to find the index of the words from any `entity_descriptor` entities from the given text. The text input is already tokenized and is given in a list form where one entry corresponds to a word or punctuation. The word indexes must be based on the list. The entities may be continuous or discontinuous, single-word or multiple words. There may also be no entities in the text.\nText: input \nReturn the output in a json format followed by a set of steps to explain how the output was generated:\n"'json [{\"entity\": entity, \"index\":[index1, index2 etc], \"type\": \"entity_type\"}, {\"entity\": entity, \"index\":[index1, index2, index3 etc], \"type\": \"entity_type\"}, etc]"'\nExplanation: explanation\n" |
| Few Shot CoT | "The task is to find the index of the words from any `entity_descriptor` entities from the given text. The text input is already tokenized and is given in a list form where one entry corresponds to a word or punctuation. The word indexes must be based on the list. The entities may be continuous or discontinuous, single-word or multiple words. There may also be no entities in the text.\n\nBelow are some examples of input text and output format.\n\nInput text: input_example_1\nExpected output: output_example_1\n\nInput text: input_example_2\nExpected output: output_example_2\n\nNow extract the entities from the text below following the examples above.\nText: input\nReturn the output in a json format followed by a set of steps to explain how the output was generated:\n"'json [\"entity\": entity, \"index\":[index1, index2 etc], \"type\": \"entity_type\", \"entity\": entity, \"index\":[index1, index2, index3 etc], \"type\": \"entity_type\", etc]"'\nExplanation: explanation\n" |

**Table E2: Variables and examples used for prompt engineering for each dataset.**

| CADEC | Value |
| --- | --- |
| entity_type | ADR |
| entity_descriptor | adverse drug reaction (ADR) |
| input_example_1 | ['Eczema', 'on', 'hands', 'and', 'feet', ',', 'rash', 'on', 'upper', 'left', 'torso', ',', 'depression', '.'] |
| output_example_1 | [{'index': [0, 1, 4], 'type': 'ADR'}, {'index': [0, 1, 2], 'type': 'ADR'}, {'index': [6, 7, 8, 9, 10], 'type': 'ADR'}, {'index': [12], 'type': 'ADR'}] |
| input_example_2 | ['My', 'fingers', 'swelled', 'up', 'and', 'hurt', '.'] |
| output_example_2 | [{'index': [1, 5], 'type': 'ADR'}, {'index': [1, 2, 3], 'type': 'ADR'}] |

| ShARe13 | Value |
| --- | --- |
| entity_type | Disorder |
| entity_descriptor | disorder |
| input_example_1 | ['1', '.', 'The', 'left', 'atrium', 'is', 'mildly', 'dilated', '.', 'No', 'atrial', 'septal', 'defect', 'is', 'seen', 'by', '2D', 'or', 'color', 'Doppler', '.'] |
| output_example_1 | [{'index': [3, 4, 7], 'type': 'Disorder'}, {'index': [10, 11, 12], 'type': 'Disorder'}] |
| input_example_2 | ['Abd', ':', 'She', 'had', 'an', 'ascitic', 'abdomen', 'that', 'was', 'very', 'large', ',', 'round', ',', 'and', 'soft', '.'] |
| output_example_2 | [{'index': [5], 'type': 'Disorder'}, {'index': [6, 15], 'type': 'Disorder'}] |

| ShARe14 | Value |
| --- | --- |
| entity_type | Disorder |
| entity_descriptor | disorder |
| input_example_1 | ['abd', 'soft', ',', 'nt', ',', 'nd'] |
| output_example_1 | [{'index': [0, 5], 'type': 'Disorder'}, {'index': [0, 3], 'type': 'Disorder'}, {'index': [0, 1], 'type': 'Disorder'}] |
| input_example_2 | ['1', '.', 'Non', '-', 'ST', '-', 'elevation', 'myocardial', 'infarction', '.'] |
| output_example_2 | [{'index': [2, 3, 4, 5, 6, 7, 8], 'type': 'Disorder'}] |

# F Case Studies

**Case 1, length: 1363 - Sample and Prompt**

"The task is to find the index of the words from any adverse drug reactions (ADR) entities from the given text. The text input is already tokenized and is given in a list form where one entry corresponds to a word or punctuation. The word indexes must be based on the list. The entities may be continuous or discontinuous, single-word or multiple words. There may also be no entities in the text.\n\nBelow are some examples of input text and output format.\n\nInput text: ['Eczema', 'on', 'hands', 'and', 'feet', ',', 'rash', 'on', 'upper', 'left', 'torso', ',', 'depression', '.']\nExpected output: [{'index': [0, 1, 4], 'type': 'ADR'}, {'index': [0, 1, 2], 'type': 'ADR'}, {'index': [6, 7, 8, 9, 10], 'type': 'ADR'}, {'index': [12], 'type': 'ADR'}]\n\nInput text: ['My', 'fingers', 'swelled', 'up', 'and', 'hurt', '.']\nExpected output: [{'index': [1, 5], 'type': 'ADR'}, {'index': [1, 2, 3], 'type': 'ADR'}]\n\nNow extract the entities from the text below following the examples above.\nText: **['Pain', 'and', 'cramping', 'in', 'my', 'hands', 'and', 'lower', 'legs', '.']**\nReturn the output in a json format followed by a set of steps to explain how the output was generated:\n```json [{\"entity\": entity, \"index\":[index1, index2 etc], \"type\": \"ADR\"}, {\"entity\": entity, \"index\":[index1, index2, index3 etc], \"type\": \"ADR\"}, etc]```\nExplanation: explanation\n"

**Case 1 - Gold Standard - 4 entities**

{'entity': 'Pain in my hands', 'index': [0, 3, 4, 5], 'type': 'ADR'},
{'entity': 'Pain in my lower legs', 'index': [0, 3, 4, 7, 8], 'type': 'ADR'},
{'entity': 'cramping in my lower legs', 'index': [2, 3, 4, 7, 8], 'type': 'ADR'},
{'entity': 'cramping in my hands', 'index': [2, 3, 4, 5], 'type': 'ADR'}

**Case 1 - Our Framework - 4 / 4 (100%)**

{'entity': 'Pain in my lower legs', 'index': [0, 3, 4, 7, 8], 'type': 'ADR'},
{'entity': 'Pain in my hands', 'index': [0, 3, 4, 5], 'type': 'ADR'},
{'entity': 'Pain in lower legs', 'index': [0, 3, 7, 8], 'type': 'ADR'},
{'entity': 'cramping in my hands', 'index': [2, 3, 4, 5], 'type': 'ADR'},
{'entity': 'cramping in my lower legs', 'index': [2, 3, 4, 7, 8], 'type': 'ADR'},
{'entity': 'cramping in lower legs', 'index': [2, 3, 7, 8], 'type': 'ADR'}

**Case 1 - W²NER - 2 / 4 (50%)**

{'entity': 'Pain in my hands', 'index': [0, 3, 4, 5], 'type': 'ADR'},
{'entity': 'cramping in my hands', 'index': [2, 3, 4, 5], 'type': 'ADR'}

**Case 1 - Gemini - Zero Shot CoT - 0 / 4 (0%)**

{'entity': 'Pain', 'index': [0], 'type': 'ADR'}, {'entity': 'cramping', 'index': [2], 'type': 'ADR'},
{'entity': 'hands', 'index': [5], 'type': 'ADR'}, {'entity': 'lower legs', 'index': [7, 8], 'type': 'ADR'}

**Case 1 - Gemini - Few Shot CoT - 0 / 4 (0%)**

{'entity': 'Pain and cramping', 'index': [0, 1, 2], 'type': 'ADR'},
{'entity': 'hands', 'index': [5], 'type': 'ADR'}, {'entity': 'lower legs', 'index': [7, 8], 'type': 'ADR'}

**Case 1 - GPT-4o - Zero Shot CoT - 0 / 4 (0%)**

{"entity": "Pain", "index": [0], "type": "ADR"}, {"entity": "cramping", "index": [2], "type": "ADR"}

**Case 1 - GPT-4o - Few Shot CoT - 0 / 4 (0%)**

{"entity": "Pain", "index": [0], "type": "ADR"},
{"entity": "cramping", "index": [2], "type": "ADR"},
{"entity": "Pain and cramping", "index": [0, 1, 2], "type": "ADR"},
{"entity": "Pain and cramping in my hands", "index": [0, 1, 2, 3, 4, 5], "type": "ADR"},
{"entity": "Pain and cramping in my hands and lower legs", "index": [0, 1, 2, 3, 4, 5, 6, 7, 8], "type": "ADR"}

**Figure F1: Case study for CADEC comparing results from trained models using our framework and a baseline and from zero and few-shot CoT prompt engineering using LLMs. The sample prompt provided follows the few-shot CoT template. All prompt templates are provided in Appendix E.**

**Case 2, length: 1572 - Sample and Prompt**

"The task is to find the index of the words from any adverse drug reactions (ADR) entities from the given text. The text input is already tokenized and is given in a list form where one entry corresponds to a word or punctuation. The word indexes must be based on the list. The entities may be continuous or discontinuous, single-word or multiple words. There may also be no entities in the text.\n\nBelow are some examples of input text and output format.\n\nInput text: ['Eczema', 'on', 'hands', 'and', 'feet', ',', 'rash', 'on', 'upper', 'left', 'torso', ',', 'depression', '.']\nExpected output: [{'index': [0, 1, 4], 'type': 'ADR'}, {'index': [0, 1, 2], 'type': 'ADR'}, {'index': [6, 7, 8, 9, 10], 'type': 'ADR'}, {'index': [12], 'type': 'ADR'}]\n\nInput text: ['My', 'fingers', 'swelled', 'up', 'and', 'hurt', '.']\nExpected output: [{'index': [1, 5], 'type': 'ADR'}, {'index': [1, 2, 3], 'type': 'ADR'}]\n\nNow extract the entities from the text below following the examples above.\nText: **['stopped', 'taking', 'it', 'after', 'almost', '2', 'years', ';', 'it', 'really', 'lowered', 'cholesterol', ',', 'but', 'at', 'cost', 'of', 'terrible', 'joint', 'pain', ',', 'loss', 'of', 'sleep', ',', 'shoulder', 'and', 'hip', 'pain', 'and', 'loss', 'of', 'range', 'of', 'motion', '.']**\nReturn the output in a json format followed by a set of steps to explain how the output was generated:\n```json [{\"entity\": entity, \"index\":[index1, index2 etc], \"type\": \"ADR\"}, {\"entity\": entity, \"index\":[index1, index2, index3 etc], \"type\": \"ADR\"}, etc]```\nExplanation: explanation\n"

**Case 2 - Gold Standard - 5 entities**

{'entity': 'joint pain', 'index': [18, 19], 'type': 'ADR'},
{'entity': 'loss of sleep', 'index': [21, 22, 23], 'type': 'ADR'},
{'entity': 'shoulder pain', 'index': [25, 28], 'type': 'ADR'},
{'entity': 'hip pain', 'index': [27, 28], 'type': 'ADR'},
{'entity': 'loss of range of motion', 'index': [30, 31, 32, 33, 34], 'type': 'ADR'}

**Case 2 - Our Framework - 5 / 5 (100%)**

{'entity': 'shoulder pain', 'index': [25, 28], 'type': 'ADR'},
{'entity': 'joint pain', {'index': [18, 19], 'type': 'ADR'},
{'entity': 'hip pain', 'index': [27, 28], 'type': 'ADR'},
{'entity': 'loss of range of motion', 'index': [30, 31, 32, 33, 34], 'type': 'ADR'},
{'entity': 'loss of sleep', 'index': [21, 22, 23], 'type': 'ADR'},
{'entity': 'terrible joint pain', 'index': [17, 18, 19], 'type': 'ADR'}

**Case 2 - W²NER - 5 / 5 (100%)**

{'entity': 'loss of sleep', 'index': [21, 22, 23], 'type': 'ADR'},
{'entity': 'hip pain', 'index': [27, 28], 'type': 'ADR'},
{'entity': 'shoulder pain', 'index': [25, 28], 'type': 'ADR'},
{'entity': 'loss of range of motion', 'index': [30, 31, 32, 33, 34], 'type': 'ADR'},
{'entity': 'joint pain', 'index': [18, 19], 'type': 'ADR'}

**Case 2 - Gemini - Zero Shot CoT - 1 / 5 (20%)**

{'entity': 'joint pain', 'index': [18, 19], 'type': 'ADR'},
{'entity': 'loss of sleep', 'index': [21, 23], 'type': 'ADR'},
{'entity': 'shoulder pain', 'index': [25, 27], 'type': 'ADR'},
{'entity': 'hip pain', 'index': [28, 30], 'type': 'ADR'},
{'entity': 'loss of range of motion', 'index': [32, 36], 'type': 'ADR'}

**Case 2 - Gemini - Few Shot CoT - 2 / 5 (40%)**

{'entity': 'joint pain', 'index': [18, 19], 'type': 'ADR'},
{'entity': 'terrible joint pain', 'index': [17, 18, 19], 'type': 'ADR'},
{'entity': 'loss of sleep', 'index': [21, 22, 23], 'type': 'ADR'},
{'entity': 'shoulder and hip pain', 'index': [24, 25, 26, 27], 'type': 'ADR'},
{'entity': 'loss of range of motion', 'index': [29, 30, 31, 32, 33], 'type': 'ADR'}

**Case 2 - GPT-4o - Zero Shot CoT - 2 / 5 (40%)**

{"entity": "terrible joint pain", "index": [17, 18, 19], "type": "ADR"},
{"entity": "loss of sleep", "index": [21, 22, 23], "type": "ADR"},
{"entity": "shoulder pain", "index": [25, 26], "type": "ADR"},
{"entity": "hip pain", "index": [27, 28], "type": "ADR"},
{"entity": "loss of range of motion", "index": [29, 30, 31, 32, 33], "type": "ADR"}

**Case 2 - GPT-4o- Few Shot CoT - 2 / 5 (40%)**

{"entity": "terrible joint pain", "index": [17, 18, 19], "type": "ADR"},
{"entity": "loss of sleep", "index": [21, 22, 23], "type": "ADR"},
{"entity": "shoulder and hip pain", "index": [25, 26, 27, 28], "type": "ADR"},
{"entity": "loss of range of motion", "index": [30, 31, 32, 33, 34], "type": "ADR"}

**Figure F2: Case study for CADEC comparing results from trained models using our framework and a baseline and from zero and few-shot CoT prompt engineering using LLMs. The sample prompt provided follows the few-shot CoT template. All prompt templates are provided in Appendix E.**

**Case 3, length: 1768 - Sample and Prompt**

"The task is to find the index of the words from any disorder entities from the given text. The text input is already tokenized and is given in a list form where one entry corresponds to a word or punctuation. The word indexes must be based on the list. The entities may be continuous or discontinuous, single-word or multiple words. There may also be no entities in the text.\n\nBelow are some examples of input text and output format.\n\nInput text: ['1', '.', 'The', 'left', 'atrium', 'is', 'mildly', 'dilated', '.', 'No', 'atrial', 'septal', 'defect', 'is', 'seen', 'by', '2D', 'or', 'color', 'Doppler', '.']\nExpected output: [{'index': [3, 4, 7], 'type': 'Disorder'}, {'index': [10, 11, 12], 'type': 'Disorder'}]\n\nInput text: ['Abd', ':', 'She', 'had', 'an', 'ascitic', 'abdomen', 'that', 'was', 'very', 'large', ',', 'round', ',', 'and', 'soft', '.']\nExpected output: [{'index': [5], 'type': 'Disorder'}, {'index': [6, 15], 'type': 'Disorder'}]\n\nNow extract the entities from the text below following the examples above.\nText: **['The', 'patient', '\"s\"', 'respiratory', 'failure', 'was', 'thought', 'to', 'be', 'secondary', 'to', 'the', 'large', 'right', 'lung', 'mass', 'as', 'well', 'as', 'the', 'copious', 'amount', 'of', 'secretions', 'that', 'she', 'was', 'producing', 'possibly', 'secondary', 'to', 'the', 'lung', 'mass', 'and', 'combination', 'of', 'her', 'underlying', 'lung', 'disease', 'of', 'chronic', 'obstructive', 'pulmonary', 'disease', '.']**\nReturn the output in a json format followed by a set of steps to explain how the output was generated:\n```json [{\"entity\": entity, \"index\":[index1, index2 etc], \"type\": \"Disorder\"}, {\"entity\": entity, \"index\":[index1, index2, index3 etc], \"type\": \"Disorder\"}, etc]```\nExplanation: explanation\n"

**Case 3 - Gold Standard - 6 entities**

{'entity': 'respiratory failure', 'index': [3, 4], 'type': 'Disorder'},
{'entity': 'lung mass', 'index': [14, 15], 'type': 'Disorder'},
{'entity': 'copious amount of secretion', 'index': [20, 21, 22, 23], 'type': 'Disorder'},
{'entity': 'lung mass', 'index': [32, 33], 'type': 'Disorder'},
{'entity': 'lung disease', 'index': [39, 40], 'type': 'Disorder'},
{'entity': 'chronic obstructive pulmonary disease', 'index': [42, 43, 44, 45], 'type': 'Disorder'}

**Case 3 - Our Framework - 6 / 6 (100%)**

{'entity': 'lung mass', 'index': [14, 15], 'type': 'Disorder'},
{'entity': 'chronic obstructive pulmonary disease', 'index': [42, 43, 44, 45], 'type': 'Disorder'},
{'entity': 'lung mass', 'index': [32, 33], 'type': 'Disorder'},
{'entity': 'lung disease', 'index': [39, 40], 'type': 'Disorder'},
{'entity': 'copious amount of secretion', 'index': [20, 21, 22, 23], 'type': 'Disorder'},
{'entity': 'respiratory failure', 'index': [3, 4], 'type': 'Disorder'}

**Case 3 - W²NER - 5 / 6 (83%)**

{'entity': 'respiratory failure', 'index': [3, 4], 'type': 'Disorder'},
{'entity': 'chronic obstructive pulmonary disease', 'index': [42, 43, 44, 45], 'type': 'Disorder'},
{'entity': 'lung mass', 'index': [14, 15], 'type': 'Disorder'},
{'entity': 'lung mass', 'index': [32, 33], 'type': 'Disorder'},
{'entity': 'lung disease', 'index': [39, 40], 'type': 'Disorder'}

**Case 3 - Gemini - Zero Shot CoT - 2/6 (33%)**

{'entity': 'respiratory failure', 'index': [3, 4], 'type': 'Disorder'},
{'entity': 'lung mass', 'index': [14, 15], 'type': 'Disorder'},
{'entity': 'lung disease', 'index': [40, 41], 'type': 'Disorder'},
{'entity': 'chronic obstructive pulmonary disease', 'index': [42, 46], 'type': 'Disorder'}

**Case 3 - Gemini - Few Shot CoT - 3/6 (50%)**

{'entity': 'respiratory failure', 'index': [3, 4], 'type': 'Disorder'},
{'entity': 'lung mass', 'index': [14, 15], 'type': 'Disorder'},
{'entity': 'secretions', 'index': [22], 'type': 'Disorder'},
{'entity': 'lung mass', 'index': [32, 33], 'type': 'Disorder'},
{'entity': 'lung disease', 'index': [37, 38], 'type': 'Disorder'},
{'entity': 'chronic obstructive pulmonary disease', 'index': [40, 41, 42, 43], 'type': 'Disorder'}]

**Case 3 - GPT-4o - Zero Shot CoT - 1 / 6 (33%)**

{"entity": "respiratory failure", "index": [3, 4], "type": "Disorder"},
{"entity": "right lung mass", "index": [12, 13, 14], "type": "Disorder"},
{"entity": "lung disease", "index": [33, 34], "type": "Disorder"},
{"entity": "chronic obstructive pulmonary disease", "index": [36, 37, 38, 39], "type": "Disorder"}

**Case 3 - GPT-4o - Few Shot CoT - 2 / 6 (50%)**

{"entity": "respiratory failure", "index": [3, 4], "type": "Disorder"},
{"entity": "large right lung mass", "index": [12, 13, 14, 15], "type": "Disorder"},
{"entity": "copious amount of secretions", "index": [20, 21, 22, 23], "type": "Disorder"},
{"entity": "lung mass", "index": [31, 32], "type": "Disorder"},
{"entity": "underlying lung disease", "index": [35, 36, 37], "type": "Disorder"},
{"entity": "chronic obstructive pulmonary disease", "index": [39, 40, 41, 42], "type": "Disorder"}

**Figure F3: Case study for ShARe13 comparing results from trained models using our framework and a baseline and from zero and few-shot CoT prompt engineering using LLMs. The sample prompt provided follows the few-shot CoT template. All prompt templates are provided in Appendix E.**

**Case 4, length: 1417- Sample and Prompt**

"The task is to find the index of the words from any disorder entities from the given text. The text input is already tokenized and is given in a list form where one entry corresponds to a word or punctuation. The word indexes must be based on the list. The entities may be continuous or discontinuous, single-word or multiple words. There may also be no entities in the text.\n\nBelow are some examples of input text and output format.\n\nInput text: ['abd', 'soft', ',', 'nt', ',', 'nd']\nExpected output: [{'index': [0, 5], 'type': 'Disorder'}, {'index': [0, 3], 'type': 'Disorder'}, {'index': [0, 1], 'type': 'Disorder'}]\n\nInput text: ['1', '.', 'Non', '-', 'ST', '-', 'elevation', 'myocardial', 'infarction', '.']\nExpected output: [{'index': [2, 3, 4, 5, 6, 7, 8], 'type': 'Disorder'}]\n\nNow extract the entities from the text below following the examples above.\nText: **['Admitted', 'to', 'the', 'pre', '-', 'op', 'holding', 'area', 'on', '[', '*', '*', '2020', '-', '05', '-', '25***', ']', ',', 'taken', 'to', 'the', 'OR', ',', 'underwent', 'CABG', 'X', '3', ',', 'PFO', 'closure', ',', 'MV', 'repair', '.']**\nReturn the output in a json format followed by a set of steps to explain how the output was generated:\n```json [{\"entity\": entity, \"index\":[index1, index2 etc], \"type\": \"Disorder\"}, {\"entity\": entity, \"index\":[index1, index2, index3 etc], \"type\": \"Disorder\"}, etc]```\nExplanation: explanation\n",

**Case 4 - Gold Standard - 1 entity**

{'entity': 'PFO', 'index': [29], 'type': 'Disorder'}

**Case 4 - Our Framework - 1 / 1 (100%)**

{'entity': 'PFO', 'index': [29], 'type': 'Disorder'}

**Case 4 - W²NER - 0 / 1 (0%)**

{'entity': 'MV', 'index': [32], 'type': 'Disorder'}

**Case 4 - Gemini - Zero Shot CoT - 0 / 1 (0%)**

{'entity': 'CABG', 'index': [22], 'type': 'Disorder'}, {'entity': 'PFO', 'index': [25], 'type': 'Disorder'},
{'entity': 'MV', 'index': [27], 'type': 'Disorder'}

**Case 4 - Gemini - Few Shot CoT - 0 / 1 (0%)**

{'entity': 'CABG', 'index': [21], 'type': 'Disorder'},
{'entity': 'PFO closure', 'index': [23, 24], 'type': 'Disorder'},
{'entity': 'MV repair', 'index': [25, 26], 'type': 'Disorder'}

**Case 4 - GPT-4o - Zero Shot CoT - 0 / 1 (0%)**

{"entity": "CABG X 3", "index": [24, 25, 26], "type": "Disorder"},
{"entity": "PFO closure", "index": [28, 29], "type": "Disorder"},
{"entity": "MV repair", "index": [31, 32], "type": "Disorder"}

**Case 4 - GPT-4o - Few Shot CoT - 0 / 1 (0%)**

{"entity": ", underwent", "index": [23, 24], "type": "Disorder"},
{"entity": "X3", "index": [26, 27], "type": "Disorder"},
{"entity": "PFO closure", "index": [29, 30], "type": "Disorder"}

**Figure F4: Case study for ShARe14 comparing results from trained models using our framework and a baseline and from zero and few-shot CoT prompt engineering using LLMs. The sample prompt provided follows the few-shot CoT template. All prompt templates are provided in Appendix E.**

Received 20 February 2007; revised 12 March 2009; accepted 5 June 2009

