# OpenReview forum: "TriG-NER: Triplet-Grid Framework for Discontinuous Named Entity Recognition"
_ACM.org/TheWebConf/2025/Conference — WWW 2025 Poster_

### Official Review · Reviewer_qnoN · 2024-11-12

**Novelty:** 5
**Technical Quality:** 5

**Review:**

This paper presents an approach to Discontinuous Named Entity Recognition (DNER), where entities may be scattered across multiple non-adjacent tokens, making traditional sequence labelling approaches inadequate. The approach, called TriG-NER is based on the introduction of a token-level triplet loss, where the word pairs belonging to an entity are pulled together in the representation space while negative word-pairs are pushed away. The approach is compared against different baselines across several datasets, consistently outperforming the baselines. The experimentation considers multiple perspectives that make the analysis rather comprehensive.

**Questions:**

In section 3.2, can you clarify the role of the convolution layer after the biLSTM? At this point, it appears to me as an unnecessary complication in the model architecture

**Reviewer Confidence:**

3: The reviewer is confident but not certain that the evaluation is correct

**Scope:**

3: The work is somewhat relevant to the Web and to the track, and is of narrow interest to a sub-community

---

### Official Review · Reviewer_Jkyb · 2024-11-24

**Novelty:** 6
**Technical Quality:** 6

**Review:**

The paper is on the topic of discontinuous named entity recognition. The author(s) propose the framework TriG-NER for this challenging problem. The proposed architecture first encodes the words in a sentence, then it builds a word pair relationship grid and applies some neural operations, and, subsequently, it applies a grid tagging and decoding scheme which classifies word-pair relationships using
three appropriate tag classes. In the final step, the framework uses a grid-based triplet loss to enables the model to learn distinctions between similar and dissimilar word pairs. The triplet loss is added to the task loss in the overall system.

Quality and clarity: the paper is well-written and clear and contains nice figures that illustrate the basic concepts very well.

Originality: the approach of the paper is original.

Significance: the results of the paper are significant given that discontinuous entity recognition is a hard problem.

Strengths:

S1. Interesting and challenging problem.

S2. Original approach

S3. Well-written paper

S4. Detailed evaluation including with experiments with large language models GPT4o and Gemini.

Weaknesses: None

**Questions:**

Q1. Please discuss what one could do for improving the performance of large language models such as GPT4o and Gemini on the task at hand.

**Reviewer Confidence:**

3: The reviewer is confident but not certain that the evaluation is correct

**Scope:**

4: The work is relevant to the Web and to the track, and is of broad interest to the community

---

### Official Review · Reviewer_DATg · 2024-12-02

**Novelty:** 4
**Technical Quality:** 5

**Review:**

This paper focuses on discontinuous named entity recognition, and proposes the triplet-grid framework TriG-NER to learn robust token-level representations for discontinuous entity extraction. The authors evaluate TriG-NER on three datasets and demonstrate significant improvements over existing grid-based architectures.

Pros:
1. Discontinuous named entity recognition has always been a research focus in NER.
2. The proposed TriG-NER achieves promising performance.
3. The writing is easy to follow.

Cons:
1. The advantages of TriG-NER compared to existing grid-based architectures are not fully explained.
2. No code and data available.

**Questions:**

1. Line 267-270 says "To the best of our knowledge, no existing work has applied a grid-based, token-level triplet loss for discontinuous named entity recognition, making our approach a novel contribution to this field". It would be better if the paper could make a deeper comparison.

2. The paper only conducts the qualitative analysis of LLMs. Can the authors do some quantitative analysis?

**Reviewer Confidence:**

4: The reviewer is certain that the evaluation is correct and very familiar with the relevant literature

**Scope:**

4: The work is relevant to the Web and to the track, and is of broad interest to the community

---

### Official Review · Reviewer_6TZa · 2024-12-02

**Novelty:** 5
**Technical Quality:** 6

**Review:**

The authors present in the paper an approach for discontinuous named entity recognition.
They use a word-to-word relation prediction instead of begin inside out (BIO) or BIOHD schemata.

The paper is well written but could further be improved when using a running example right from the beginning (e.g. the one in Figure 3) to understand the idea even better.
Especially when and how two words are related in the grid.

The evaluation part is fine, and a lot of ablation studies are done.
One can see that the improvements are rather marginally better over the other approaches.
What I'm missing is an analysis of the impact of different random initializations of the head of the encoder
because it can happen that the improvements are not so big anymore.
Thus, include multiple runs and provide a deviation or argue why the whole setup is deterministic.

The discussion about LLMs is fine, but if one really wants to implement an approach,
the indices and number of words (lines 803-808) could be easily fixed by a postprocessing step.
It would be interesting to see how an LLM performs instead of an encoder-only model in predicting the relations between the words
(similar task, but just replacing BERT with an LLM).

All datasets that exist for discontinuous named entity recognition are based on the medical domain.
Thus, it cannot be analyzed how the approach will perform in other domains.
However, this shows a lack of corresponding datasets.

One of the things that is missing is the link to the code and datasets to reproduce the results of the paper.


Minor points:
- In line 545, the threshold used for the early stopping is not mentioned.
- Line 562: shortly explain what exactly DiscEnt contains (is it only the span where each entity fully appears?)
- No CCS concepts where provided

**Questions:**

1) Do the authors run the experiments multiple times? What are the standard deviations, or why is the approach deterministic?
2) Is each token used as once as an anchor?
3) Where have your data and code been published?

**Reviewer Confidence:**

2: The reviewer is willing to defend the evaluation, but it is likely that the reviewer did not understand parts of the paper

**Scope:**

4: The work is relevant to the Web and to the track, and is of broad interest to the community

---

### Official Review · Reviewer_4DwX · 2024-12-03

**Novelty:** 4
**Technical Quality:** 4

**Review:**

Pros
1.  Clear structure and good writing.
2.  Significant performance improvement on three benchmark DNER datasets.
3.  TriG-NER is adaptable to different datasets and entity types.

Cons
1.  The main innovations of TriG-NER overlap with some related works in the field, but the paper does not discuss this.
2.  The paper does not provide comparison results with open-source or closed-source models like GPT-4, Gemini, Llama 3, etc.

**Questions:**

1.  Please provide comparison results with models like GPT-4, Gemini, Llama 3, etc.
2.  References [1] and [2] also propose Triplet Loss, while TriG-NER uses almost the same L_{triplet} and defines several Selection categories. Please explain the differences and comparisons between TriG-NER and these two papers.

[1] Multi-Task Triplet Loss for Named Entity Recognition using Supplementary Text
[2] Meta-Learning Triplet Network with Adaptive Margins for Few-Shot Named Entity Recognition

**Reviewer Confidence:**

2: The reviewer is willing to defend the evaluation, but it is likely that the reviewer did not understand parts of the paper

**Scope:**

4: The work is relevant to the Web and to the track, and is of broad interest to the community